# Plant-Derived Quorum Sensing Inhibitors (Quercetin, Vanillin and Umbelliferon) Modulate Cecal Microbiome, Reduces Inflammation and Affect Production Efficiency in Broiler Chickens

**DOI:** 10.3390/microorganisms11051326

**Published:** 2023-05-18

**Authors:** Dmitry G. Deryabin, Dianna B. Kosyan, Ksenia S. Inchagova, Galimzhan K. Duskaev

**Affiliations:** Federal Research Centre of Biological Systems and Agro-Technologies of the Russian Academy of Sciences, 460000 Orenburg, Russia; dkosyan74@gmail.com (D.B.K.); ksenia.inchagova@mail.ru (K.S.I.); gduskaev@mail.ru (G.K.D.)

**Keywords:** quorum sensing inhibitors, quercetin, vanillin, umbelliferon, cecal microbiome, broiler chickens

## Abstract

Quorum sensing inhibitors (QSIs) are an attractive alternative to antibiotic growth promoters in farmed animal nutrition. The goal of the study was the diet supplementation of Arbor Acres chickens with quercetin (QC), vanillin (VN), and umbelliferon (UF), which are plant-derived QSIs preliminarily showing cumulative bioactivity. Chick cecal microbiomes were analyzed by 16s rRNA sequencing, inflammation status was assessed by blood sample analyses, and zootechnical data were summarized in the European Production Efficiency Factor (EPEF). When compared to the basal diet control group, a significant increase in the *Bacillota:Bacteroidota* ratio in the cecal microbiome was found in all experimental subgroups, with the highest expression > 10 at VN + UV supplementation. Bacterial community structure in all experimental subgroups was enriched with *Lactobacillaceae* genera and also changed in the abundance of some clostridial genera. Indices of richness, alpha diversity, and evenness of the chick microbiomes tended to increase after dietary supplementation. The peripheral blood leukocyte content decreased by 27.9–45.1% in all experimental subgroups, likely due to inflammatory response reduction following beneficial changes in the cecal microbiome. The EPEF calculation showed increased values in VN, QC + UF, and, especially, VN + UF subgroups because of effective feed conversion, low mortality, and broiler weight daily gain.

## 1. Introduction

Since the middle of the 20th century, low doses of antimicrobials known as antibiotic growth promoters (AGPs) have been used for diet supplementation in broiler chickens. It was considered that AGPs suppress zoopathogenic bacteria in the gut microbiome, which reduces the inflammatory response and improves feed conversion rates, maximizing the growth potential of farm animals [1]. However, in 2000 (following a World Health Organization study confirming the negative effects of AGPs), it was recommended to drastically reduce or stop the use of antimicrobials in livestock production. On 1 January 2006, the European Union first introduced a ban on the use of four types of antimicrobials as growth promoters in animal feed [2], and on 28 January 2022, completely prohibited the regular feeding of farm animals with antimicrobials [3].

The AGPs ban has encouraged researchers to find alternative antibiotic-free strategies for gut and overall health in broiler chickens, including prebiotics, probiotics, synbiotics, bacteriophages, and vaccines [4]. In this concept, a very attractive perspective is offered by novel bioactivity compounds targeted to cell-to-cell communication in bacteria named quorum sensing inhibitors (QSIs) [5]. Briefly, many bacterial species synchronize differential gene expression via the production, release, and reception of small chemical signaling molecules called autoinducers (AIs), whose environmental concentration is proportional to cell population density [6]. Due to the fact that in zoopathogenic bacteria this phenomenon regulates virulence traits and biofilm formation, quorum sensing has been evaluated as a promising antibacterial target where QSIs provide effective reduction of pathogenic behaviors without significant selective pressure against the bacterial population [7]. Remarkably, natural QSIs are often plant-derived compounds found in edible or medicinal plants [8], which gives a chance for a rapid transition from scientific research to practical use in livestock as an alternative to AGPs. For example, the unsaturated bicyclic monoterpene α-pinene, which is the major component of conifer extracts and also found in many non-coniferous plants—camphor weed (*Heterotheca*) and great sagebrush (*Artemisia tridentata*), effectively inhibits quorum sensing in *Campylobacter jejuni* and reduces colonization by this widespread foodborne zoopathogen in broiler chickens [9].

Our attention was focused on three groups of plant-derived compounds (flavonoids, phenolic aldehydes, and coumarins) that exhibit significant QSI effects and well-documented bioactivity mechanisms, have low toxicity to animals, and are approved for use in food supplementation. Thus, the flavonoid compound quercetin (QC), found in many fruits, vegetables, leaves, seeds, and grains, shows a significant reduction in QS-regulated phenotypes in foodborne bacteria via binding AIs receptors [10], and on the other hand, the FDA recognizes high-purity quercetin for use as an ingredient in various food categories [11]. Another plant-derived compound is the phenolic aldehyde vanillin (VN), which was first characterized as QSI in vanilla bean extract (*Vanilla planifolia Andrews*) [12] and, based on transcriptomic data, suppresses mobility, adhesion, chemotaxis, extracellular polymer substance secretion, and AI release in bacteria [13]. In the current context, it is significant that, according to the FDA recommendation, not only natural vanilla extract but also synthetic vanillin is allowed as a flavoring agent in foods, beverages, and pharmaceuticals [14]. Another attractive compound is umbelliferon (UF), which belongs to the coumarin group and is found in many edible plants from the *Apiaceae* family, such as carrot, coriander, and garden angelica. In turn, proteomic profiling revealed the UF-induced downregulation of major virulence-associated proteins and QS-related transcription factors in bacteria [15]. In the preliminary stage of this study, we tested QC + UF and VN + UF compositions in the *Chromobacterium subtsugae* (formerly *C. violaceum*) bioassay [16], which showed a cumulative QSI effect, justifying the combined use of these compounds in diet supplementation.

Notably, QC, VN, and UF show a variety of biological activities, which are not limited to microorganism effects but also include host effects. The antioxidant, anti-inflammatory, wound-healing, anti-diabetic, and cardiovascular properties of QC have been extensively researched [17]. Recently, the bioactive properties of VN, such as antioxidant, neuroprotective, and anti-carcinogenic effects, have attracted increasing attention [18]. UV exhibits anti-inflammatory activity as well as antihyperglycemic, molluscicidal, and antitumor properties [19]. Some of these bioactivities may be useful in animal husbandry, further supporting the introduction of the considered plant-derived compounds in farmed poultry feeding.

The available literature contains several reports on the use of QC, VN, and UF in poultry nutrition. Some nutritional and beneficial effects have been reported for QC, where supplementation can lead to a state of immune alertness and a lower incidence of infectious diseases [20]. The QC addition in the broiler chick diet also had a positive effect on dry matter digestibility and energy retention, as well as improved breast muscle yield and quality [21], protecting it against lipid oxidation and deposition by regulating the PI3K/PKB/AMPKα1 signaling pathway [22]. Diet supplementation with a microencapsulated blend of citric and sorbic acids, thymol, and VN increased the functional activity of peripheral blood leukocytes in broiler chickens [23] and also led to a decrease in intramuscular fat content and an overall improvement in the fatty acid profile in chicken meat [24]. Reported UF effects include prevention of oxidative stress, inflammation, and hematological alterations [25], as well as antidiarrheal and antiulcerogenic activity [26]. However, its usage in poultry feeding has not yet been demonstrated. At the same time, there are only a few publications on QC, VL, and UF in the AGPs paradigm, including their impact on the gut microbiome, inflammatory responses, and productive performance of broiler chicken [27,28,29], which does not yet prove them as AGPs alternatives in livestock.

The goal of the study was the evaluation of plant-derived quorum sensing inhibitors quercetin, vanillin, and umbelliferon supplemented with a basal diet in broiler chickens, focusing on the cecal microbiome, blood inflammation status, and growth performance of poultry.

## 2. Materials and Methods

### 2.1. Study Design

The study was performed in accordance with the European Convention for the Protection of Vertebrate Animals used for Experimental and other Scientific Purposes (18 March 1986) and the principles of good laboratory practice. The study protocol was approved by the Animal Ethics Committee of the Federal Scientific Center for Biological Systems and Agro-Technologies of the Russian Academy of Sciences.

A total of 450 Arbor Acres chicks (Aviagen LLC, Tula, Russia), which reached 7 days’ age at the start of the experiment, were randomly separated into 5 groups of 90 birds and then divided into three trials of 30 animals each. The chickens were allocated into cages of 0.9 × 0.45 m (5 birds per cage) and fed a basal diet (BD) in accordance with the Arbor Acres broiler nutritional recommendations [30]. The control group received the BD only, while the feed for the experimental groups was supplemented with QC (10 mg/kg daily), QC + UF (2.5 mg/kg and 0.1 mg/kg per day, respectively), VN (0.5 mg/kg daily), or VN + UF composition (0.3 mg/kg each per day). The dosage calculation in the QC + UF and VN + UF compositions was based on preliminary in vitro experiments in which a QSI effect similar to that of QC or VN alone was achieved, respectively. All these compounds, quercetin dehydrate (C15H10O7 × 2H_2_O; CAS 6151-25-3), vanillin (C8H8O3; CAS 121-33-5), and umbelliferon (C9H6O3; CAS 93-35-6), were purchased from Acros Organics BVBA (Belgium) with a purity of 99%.

At day zero and days 7, 14, 21, and 35 after diet supplementation, the survival rate, weight gain, and feed intake were recorded in each group. Finally, the European Production Efficiency Factor was calculated on the basis of robust zootechnical data. At the end of the experiment, 10 chickens were randomly selected from each group, and they had individual blood samples taken from a wing vein into a 2 mL EDTA tube and promptly processed to assess general inflammation status. After that, the same birds were humanely euthanized, and the 200 µL of each cecum’s contents were massaged into individual sterile Eppendorf tubes containing 200 µL of DNA/RNA Shield (Zymo Research, Irvine, CA, USA), immediately frozen on dry ice, and stored at −80 °C until DNA extraction and following 16S-rRNA gene sequencing.

The principal study design is shown in Figure 1.

### 2.2. Cecal Microbiome DNA Extraction, Sequencing and Data Analysis

Total DNA from each ceca sample was extracted with a commercial QIAamp Fast DNA Stool MiniKit (Qiagen GmbH, Hilden, Germany) according to the manufacturer’s instructions. DNA concentration was determined on a Qubit 4 fluorometer (Life Technologies, Carlsbad, CA, USA) using the dsDNA high-sensitivity assay kit (Life Technologies, USA). The DNA quality was evaluated by 1% agarose gel electrophoresis.

16S-rRNA gene libraries were prepared according to the two-stage Illumina protocol (Part #15044223, Rev. B). The V3–V4 variable regions of this gene were amplified using the forward S-D-Bact-0341-b-S-17 (CCTACGGGNGGCWGCAG) and the reverse S-D-Bact-0785-a-A-21 (GACTACHVGGGTATCTAATCC) primers containing the overlapping region of the Illumina sequencing primers [31]. The DNA libraries were validated by real-time PCR on CFX Connect (BioRad, Hercules, CA, USA). High-throughput paired-end 2 × 250 bp sequencing was performed on the Illumina MiSeq platform with the V.2 reagent kit (Illumina, San Diego, CA, USA) for 500 cycles.

Bioinformatic analysis of raw data was performed using the USEARCH V.10.0.240 software following the operational protocol recommended by R. Edgar, which included merging of reads into contigs, filtering of contigs by length (at least 420 bp) and quality (maxee 1.0), chimera deletion, dereplication, and clustering into separate operational taxonomic units (OTU) [32]. After data clean up, the final OTU taxonomic affiliation at the phylum, class, order, family, and genus level was carried out using the RDP Classifier algorithm against the SILVA database [33].

### 2.3. Blood Samples Analyses

The blood cell count was performed using an URIT-2900Vet Plus hematological analyzer for veterinary use (URIT Medial Electronic Co., Guilin, China) and manufacturer’s reagents based on Coulter’s electrical resistance measurement method to obtain the absolute values of white blood cells (WBC), red blood cells (RBC), and platelets (PLT), which were presented as the number of cells in 1 L of blood. In addition, a differential counting of 3 WBC types (lymphocytes, monocytes, and granulocytes) was performed, which gave their relative values (%). Hemoglobin, hematocrit, and some other blood parameters expressed in g/L and % values were analyzed on the same device using the photometric method.

### 2.4. Zootechnical Data

Feed conversion was evaluated based on the weight gain and feed intake in each broiler chicken group. To compare technical results, taking into account feed conversion, mortality, and daily gain, a standardized European production efficiency factor (EPEF) was calculated according to the formula:EPEF = (average growth per day × survival rate)/feed conversion × 10,
where an average growth is calculated as final weight gain in grams divided by 35 days of feeding, and a survival rate is calculated as 100—% mortality.

### 2.5. Statistical Analysis

The relative abundances of the main phyla and the *Bacillota:Bacteriodota* (Bc:Bd) ratios found in each group were represented by column charts. The estimated abundance of each of the top 20 genera exceeded 1 percent in the microbiome and represents the median value. Differences in the phyla and genera abundances in the control and experimental groups were assessed by the Mann–Whitney (U) test. Microbiome richness, alpha diversity, and evenness were estimated by the Chao-1, Margalef, Dominance D, Shannon, and Simpson (1-D), and evenness eH/S indexes were calculated using PAST V. 4.06 software [34]. The blood counting results, as well as zootechnical data management, were performed using the SPSS Statistics 20.0 software (IBM, Armonk, NY, USA), and the mean (M), standard deviations (±σ), standard deviation errors (±SE) were calculated. *p* values < 0.05 were considered significant.

## 3. Results

### 3.1. Effect of QC, VN and UF Diet Supplementation on Ceca Bacterial Community Composition in Broiler Chickens

In order to determine which bacterial taxa were present in the cecal microbiome, the bacterial communities were analyzed by 16s rRNA sequencing, and the detected OTUs were affiliated at the phylum, class, order, family, and genus levels.

At the phylum level, four bacterial taxa were identified within each control and experimental group: *Actinomycetota*, *Bacillota*, *Bacteroidota,* and *Mycoplasmatota* (formerly known as *Actinobacteria*, *Firmicutes*, *Bacteroidetes,* and *Tenericutes,* respectively) [35]. Members of the *Bacillota* and *Bacteroidota* phylas made up the majority of the chicken’s gut microbiome. The phylum *Bacillota* was represented by 51 genera and unclassified members of the *Bacillales* or *Clostridiales* classes, while the phylum *Bacteroidota* included only one *Bacteroides* genus. About 1% of OTUs were represented by *Candidatus Melainabacteria*. Unclassified OTUs in the *Bacteria* domain were less than 0.01%. OTUs belonging to the phylum *Pseudomonadota* (formerly *Proteobacteria*) were not found.

The phylum structure of bacterial communities is shown in Figure 2.

The Top-20 most abundant bacterial genera in the microbiomes of the control and experimental groups are presented in Table 1.

The diversity indexes integrally assessing the microbial community of the chicken ceca and calculated using PAST3 are presented in Table 2.

In the control group microbiome, members of the *Bacillota* and *Bacteroidota* phyla showed approximately equal presence, resulting in a Bc:Bd ratio of 1.195. The most predominant genus was *Bacteroides* (44.95%), while other abundant bacterial genera were members of the class *Clostridiales: Agathobaculum, Anaerotignum, Eisenbergiella, Faecalibacterium, Romboutsia,* as well as the genus *Turicibacter* belonging to the class *Erysipelotrichia*, each representing 1.06–2.86% of the bacterial community (Table 1). In sum, the control group microbiome consisted of 55 taxa (Chao-1 index = 55), and the Margalef index was 5.334, indicating a high level of taxa richness. The dominance D index calculation gave a 0.2629 value that reflects a significant abundance of *Bacteroides* genus members. Shannon index values of 2.079 and Simpson (1 − D) index values of 0.7371 indicated moderate alpha diversity. The evenness eH/S index value of 0.1453 showed that there was no evenness in the control gut microbial community (Table 2).

The diet supplementation with plant-derived compounds significantly modulated the chicken’s cecal microbiome, which manifested itself in a change in the main phyla ratios, a decrease or increase in some genera abundance, and was reflected in the biodiversity index values. The general signs of the microbial community’s changes were found, as well as specific changes that form only when QC, VN, or UF are used.

The common change following the diet supplementation was a decrease in the *Bacteroidota* phyla abundance with a simultaneous increase in the *Bacillota* phyla abundance, which enhanced the Bc:Bd ratio values (Figure 2). This was least pronounced in the QC experimental subgroup (Bc:Bd = 1.613), manifested itself with the use of VN or QC + UF composition (Bc:Bd ratios of 3.146 and 4.128, respectively), and was most significant in the VN + UF subgroup microbiome. The VN + UF diet supplementation led to a 5-fold decrease in *Bacteroidota* phyla abundance (8.44% vs. 44.95% in the control group; *p* < 0.001), while *Bacillota* phyla abundance increased from 53.73% to 90.10% (*p* < 0.01), resulting in a Bc:Bd ratio of 10.726. The minor phyla abundance in the chicken’s gut microbiome did not significantly change due to the diet supplementation.

The Top-20 bacterial genera comparative analysis showed an increase in the abundance of taxa belonging to the family *Lactobacillaceae*, order *Lactobacillales*, and class *Bacilli*. Against the control group, the bacterial community structure in all experimental subgroups was 3,4-30,3-fold enriched with *Lactobacillus*, 4,4-23,2-fold with *Ligilactobacillus*, and 7,3-57,7-fold with *Limosilactobacillus* genera (*p* < 0.05). Other common changes included an enrichment in the genera *Christensenella* (except the QC + UF subgroup) and *Mediterraneibacter* (except the VN subgroup). Specific microbiome changes depending on the diet supplementation composition were related to the class *Clostridia*: enrichment of the *Anaerotignum* genus when QC was used, high abundances of the *Faecalibacterium* genus when UF was included in feed compositions, progression of the *Subdoligranulum* genus in the VN + UF supplemented group, etc. The *Bacteroides* genus abundance was similar to that of the *Bacteroidota* phyla (see above) and decreased significantly (*p* < 0.05) in all experimental groups.

The richness, alpha diversity, and evenness index calculations showed common tendencies in the chicken’s microbiomes following the diet supplementation (Table 2). An upward trend in the genera numbers in the experimental group microbiomes was revealed (Chao-1 values grew up from 55 to 64–65), and the Margalef index values were set at 6.124–6.404 vs. 5.334 of the control value, which indicated the progression of taxa richness. The Dominance_D index values decreased from 0.2629 to 0.1264–0.2001 due to a decrease in the *Bacteroides* genus abundance, which dominates in the control group while no taxon dominates in the experimental groups in the bacterial community. The Shannon index calculation gave values of 2.323–2.733 (against a 2.079 value in the control group) and Simpson (1 − D) values of 0.7999–0.8736 (against a 0.7371 value in the control group), which means that the alpha diversity has increased and varied from moderate to high. The evenness eH/S index increased to 0.1906, 0.2195, and 0.2365 in VN + UF, QC + UF, and VN subgroups (against a 0.1453 value in the control group), indicating ongoing unevenness in the microbial community. Specific changes in cecal microbiome biodiversity depending on diet supplementation have not been found.

### 3.2. Effect of QC, VN and UF Diet Supplementation on Hematological Parameters in Broiler Chickens

The blood parameters were measured at the end of the experiment in 42-day-old chicks after 35 days of diet supplementation. The values of 15 parameters compared between the control and experimental groups are presented in Table 3.

The white blood cell (WBC) count was decreased in all experimental groups (*p* < 0.05) and most significantly after VN + UF supplementation (27.23 ± 1.51 × 10^9^ cell/l against 49.60 ± 8.94 × 10^9^ cell/L in the control; *p* < 0.001). The WBC ratio calculation showed that the total peripheral blood leukocyte content decreased by 27.9–45.1% after diet supplementation, while the ratio between WBC types did not change and the relative values of lymphocytes (LYM), monocytes (MONO), and granulocytes (GRAN) remained stable.

The diet supplementation did not affect the red blood cell (RBC) count, as well as RBC-related qualitative parameters: hemoglobin (HGB), hematocrit (HTC), erythrocyte mean corpuscular volume (MCV), mean concentration of hemoglobin (MCH), mean corpuscular hemoglobin concentration (MCHC), and red cell distribution (RDW).

The data analyses also showed changes in the platelet (PLT) count following QC, QC + UF, and VN diet supplementation (62.00 ± 5.28 × 10^9^ cell/L, 61.75 ± 4.61 × 10^9^ cell/L, and 65.00 ± 5.58 × 10^9^ cell/L against the control value of 91.00 ± 19.85 × 10^9^ cell/L; *p* < 0.05), while the thrombocrit (PTC) level was decreased in all experimental groups without statistically significant differences in the mean platelet volume (MPV).

### 3.3. Effect of Diet Supplementation on the Broiler Chickens Zootechnical Characteristics

Throughout the experiment, the chicken’s survival rate was 94% in the control group, increasing to 96% in the QC supplemented subgroup and up to 98% in the QC + UF, VN, and VN + UF subgroups.

The final broiler weight was 2736.50 ± 215.4 g in the control group, which, taking into account the initial weight of 7-day-old chickens equal to 182.50 ± 8.1 g, gave an average growth per day of 72.97 g. Diet supplementation with QC and QC + UV did not change these zootechnical data, while in the VN group, the weight parameters tended to increase to 2816.00 ± 135.7/75.24 g, being significantly higher compared with the control after VN + UV supplementation (2847.00 ± 160.4/76.13 g; *p* < 0.05).

The feed conversion calculation based on broiler chickens’ weight gain and total starter and finisher intake was 1.653 g in the control group, was similar in the QC and VN groups, and decreased to 1.585 g and 1.572 g in the QC + UF and VN + UV groups, respectively.

For an integral comparison of zootechnical results, a standardized European Production Efficiency Factor (EPEF) was calculated based on the poultry survival rate, daily gain, and feed conversion (Figure 3). The EPEF value was 414.89 in the control group, decreased to 395.95 after QC supplementation, and increased in the VN, QC + UF, and, especially, VN + UF groups: 450.51, 449.02, and 474.55, respectively.

## 4. Discussion

This study aimed to analyze plant-derived quorum sensing inhibitors (quercetin, vanillin, and umbelliferon) as an alternative to antibiotic growth promoters in poultry nutrition. In this paradigm, QC, VN, and UF dietary supplementation were predicted to have beneficial effects on the broiler’s cecal microbiome composition, suppress inflammation, and enhance nutrient availability, which together should lead to overall health and growth efficiency [36].

Based on 16s rRNA sequencing data, the major bacterial phyla identified in the chicken cecal microbiome were *Actinomycetota*, *Bacillota*, *Bacteroidota,* and *Mycoplasmatota*. Surprisingly, members of the *Pseudomonadota* phylum, including the most important foodborne pathogens *Escherichia*, *Salmonella,* and *Campylobacter*, have not been found. Since these zoopathogens have well-characterized quorum sensing networks used to induce virulence traits and biofilm development, they are considered a promising target for QSI [37]. However, due to their absence in the microbiome, this hypothesis could not be verified, and the used Arbor Acres chickens were rated as specific pathogen-free lines.

Against this background, *Bacillota* and *Bacteroidota* bacterial phyla were the most dominant, while QSI diet supplementation stereotypically reduced the number of *Bacteroidota* phyla but increased the relative abundance of *Bacillota* phyla. Compared to the control group, which showed an approximately equal abundance of these phyla, the *Bc*:*Bd* calculation gave increased values of this ratio in all experimental groups, with the most expressed >10 with the VN + UF supplementation. In this regard, a recent report on quorum sensing in *Bacteroidota* phyla members [38] should be referred to, which may explain the QSI’s impact on the chicken’s gut microbiome. Another explanation could be an inhibitory effect of similar plant-derived molecules on *Bacteroides* lactate dehydrogenase [39] or a disruption of the starch utilization system, depriving these bacteria of their primary metabolic energy source [40]. However, this assumption requires further evidence regarding the QC, VN, and UF effects in *Bacteroides* spp.

In this study, it was observed that diet supplementation leads to a significant increase in the relative abundance of the *Bacillota* phylum, especially from the *Lactobacillaceae* family, including *Lactobacillus*, *Ligilactobacillus,* and *Limosilactobacillus* genera. This finding is in good agreement with known data on the effects of QC diet supplementation on broiler chickens’ microbiomes [21,27] and should be evaluated as potential gut microbiota-associated benefits. Through efficient carbohydrate fermentation, *Lactobacillaceae* provide significant assistance to host metabolism, improving feed conversion rates and reducing broiler mortality [41]. Another beneficial effect of QC supplementation was the enrichment of the cecal microbiome with the *Anaerotignum* genus, which belongs to the short-chain fatty acid producing family *Lachnospiraceae* [42]. The present study demonstrates a similar modulation following VN supplementation and also shows for the first time UF-specific enrichment of the chicken’s microbiome with the genus *Faecalibacterium*, which is a prominent butyrate producer important for gut epithelial health [43] and may be related to improved broiler growth performance [44]. This finding additionally showed QC, VN, and UF as prebiotic-like bioactive molecules, and the resulting increase in the abundance of the *Bacillota* taxa mentioned above should be viewed as a positive diet supplementation outcome.

Overall, indices of richness, alpha diversity, and evenness of the chick microbiomes tended to increase after dietary supplementation, which are scalable metrics of the intestinal health of poultry [45]. In sum, these changes tend to increase the range of processed substrates and lead to an expansion of the metabolic potential of the cecal microbiome, which positively correlates with improved feed conversion ratio and feed efficiency [46].

The QC, VN, and UF diet supplementation also led to inflammatory response reduction as assessed by broiler peripheral blood cell count [47]. The total leukocyte content decreased by 27.9–45.1% without changes in lymphocyte, monocyte, or granulocyte ratios. Additionally, the datasets showed a decrease in platelet count when QC, QC + UF, and VN were used, as well as in thrombocrit values in all experimental subgroups. In line with the concept of avian platelets as specialized immune cells [48], these changes can also be assessed as having an anti-inflammatory effect following diet supplementation. One of the possible reasons for this effect is beneficial changes in the cecal microbiome, in particular the decreased abundance of *Bacteroidota* phyla members, which are producers of lipopolysaccharide (LPS) [49], a cell-associated glycolipid of the outer membrane of Gram-negative bacteria, and a canonical mediator of microbe-host interactions via LPS-induced inflammation [50]. Another reason is the direct QC, VN, and UF beneficial effects on inflammation and hematological alterations due to the attenuation of oxidative stress, inhibition of LPS-induced cytokine production [51], suppression of mitogen-activated protein kinases and the nuclear factor kappa-light-chain-enhancer activation pathway [52], or modulation of glutamate-nitric oxide-cGMP signaling [25].

Finally, supplemented broilers were observed with higher EPEF averages in the VN, QC + UF, and VN + UF groups than their basal diet control counterparts, indicating improved bird health and effective productivity. This data supports the interconnection between the microbiome, gut health, and chicken productivity [53], which suggests a key role for the cecal microbiome in maintaining host homeostasis, mainly through the control of inflammation and therefore decreasing the energy expense that poultry invest in maintaining immune system activity. Notably, the highest EPEF value was recorded in the VN + UF supplemented subgroup, which had the highest Bc:Bd ratio, the lowest WBC count, and a statistically significant final broiler weight. Thus, the study results showed the combined plant-derived QSIs (vanillin + umbilliferon) diet supplementation as a promising approach to alternative antibiotic-free poultry production strategies, providing low mortality, effective feed conversion, and broiler weight daily gain through beneficial gut microbiome modulation and associated reduction of gut inflammation.

## 5. Conclusions

Plant-derived compounds with various beneficial biological activities and low toxicity are increasingly being proposed for farm animals feeding. Currently, these commercially available products are generally classified as nature-identical botanical compounds (NICs) and used in a non-antibiotic strategy to improve the health and performance of poultry [54]. In this study, our attention was focused on three NICs (quercetin, vanillin, and umbelliferon), which were preliminary evaluated as quorum sensing inhibitors with well-documented additional beneficial effects on the host. Supplementation of the basal diet in broiler chickens with these compounds singly and in combinations led to complex modulation of the cecal microbiome, reduction of inflammation, and increased production efficiency in broiler chickens, which is commonly expected as an effect of antibiotic growth promoters. Thus, the obtained data support the use of NICs in animal nutrition, showing quorum sensing inhibition as a promising way to select the most effective plant-derived compounds for antibiotic-free diet supplementation in livestock.

## Figures and Tables

**Figure 1 microorganisms-11-01326-f001:**
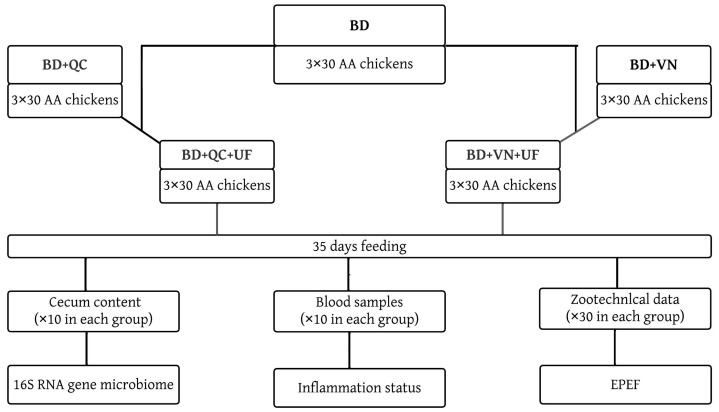
Study design for evaluation of quercetin, vanillin, and umbelliferon effects on gut microbiome, inflammation status, and growth performance of broiler chickens. Abbreviations: AA—Arbor Acres; BD—basal diet; QC—quercetin supplementation; VN—vanillin supplementation; UF—umbelliferon supplementation; EPEF—European Production Efficiency Factor.

**Figure 2 microorganisms-11-01326-f002:**
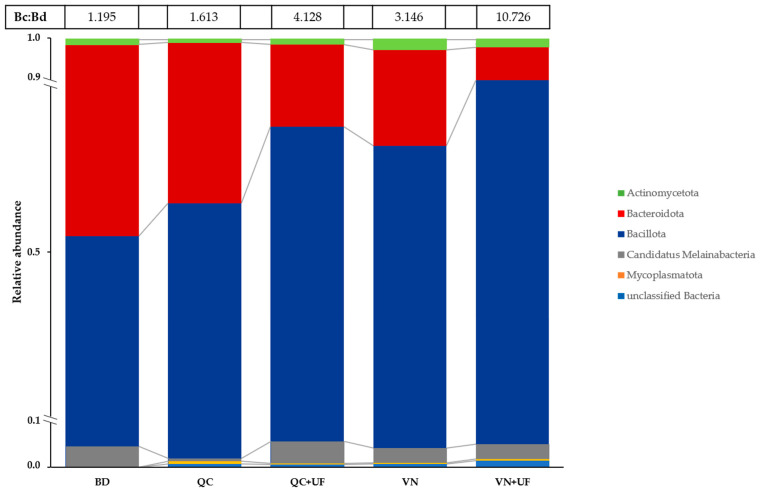
The relative abundance of the bacterial phyla in the cecal microbiomes of control and experimental groups.

**Figure 3 microorganisms-11-01326-f003:**
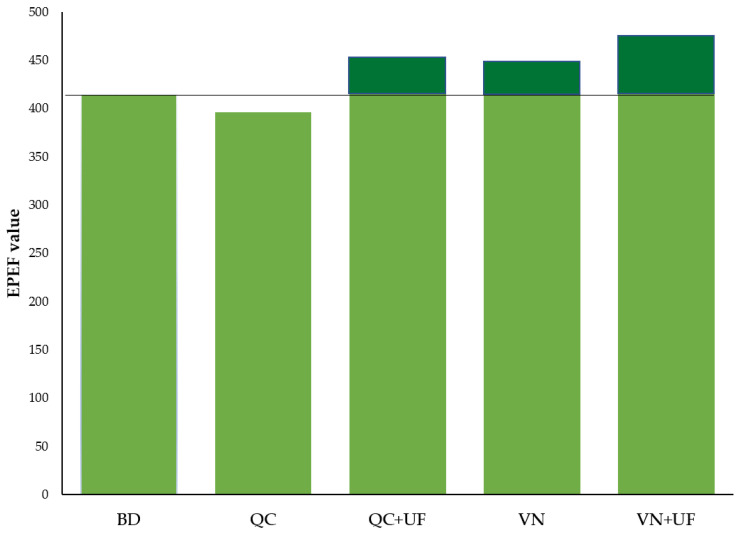
European Production Efficiency Factor values for broiler chicken groups supplemented with quercetin (QC), vanillin (VN), and umbelliferon (UF) compared to the basal diet control group.

**Table 1 microorganisms-11-01326-t001:** Top-20 genera in the chicken’s cecal microbiome.

Phylum	Class	Order	Family	Genus	Diet Supplementation Groups				
					BD	QC	QC + UF	VN	VN + UF
*Bacillota*	*Bacilli*	*Lactobacillales*	*Lactobacillaceae*	*Lactobacillus*	0.09	0.62 *	2.73 *	0.31 *	0.93 *
				*Ligilactobacillus*	0.12	0.53 *	2.29 *	2.78 *	1.02 *
				*Limosilactobacillus*	0.03	0.33 *	1.30 *	0.22 *	1.73 *
				*Streptococcus*	0.03	0.02	0.10	0.63 *	1.28 *
	*Clostridia*	*Eubacteriales*	*Christensenellaceae*	*Christensenella*	0.76	6.83 *	1.70	5.75 *	4.99 *
			*Lachnospiraceae*	*Agathobacter*	0.17	0.19	0.07	2.39 *	0.22
				*Anaerotignum*	1.74	6.89 *	1.59	0.53	0.16 *
				*Eisenbergiella*	2.68	2.74	2.03	2.61	0.45 *
				*Mediterraneibacter*	0.43	1.63 *	1.55 *	0.96	2.81 *
			*Oscillospiraceae*	*Agathobaculum*	2.62	0.95	0.29 *	0.76	0.10 *
				*Butyricicoccus*	0.59	1.10	0.68	1.36	1.00
				*Dysosmobacter*	0.95	0.59	1.87	1.64	0.97
				*Faecalibacterium*	1.87	1.65	5.25 *	2.57	31.60 *
				*Oscillibacter*	0.44	0.34	0.64	1.07	0.13
				*Ruthenibacterium*	0.85	0.26	1.12	0.30	0.07 *
				*Subdoligranulum*	0.90	0.19	0.46	1.13	6.60 *
			*Peptostreptococcaceae*	*Romboutsia*	1.10	1.05	0.39	0.63	0.07 *
	*Erysipelotrichia*	*Erysipelotrichales*	*Erysipelotrichaceae*	*Turicibacter*	1.71	0.31	1.04	1.18	0.17 *
*Bacteroidota*	*Bacteroidia*	*Bacteroidales*	*Bacteroidaceae*	*Bacteroides*	44.95	38.04 *	19.20 *	23.76 *	8.44 *
*Candidatus Melainabacteria*	*Melainabacteria*	*Vampirovibrionales*		*Vampirovibrio*	1.05	0.15 *	1.12	0.75	0.75

The values in the columns represent the median abundance (%) of the specified taxon. *—*p* < 0.05 in comparisons with the control group.

**Table 2 microorganisms-11-01326-t002:** Diversity indexes of microbial communities in the chicken’s ceca.

Indexes	Diet Supplementation Groups				
	BD	QC	QC + UF	VN	VN + UF
Chao-1	55	64	65	65	64
Margalef (Dmg)	5.334	6.124	6.403	6.404	6.263
Dominance D	0.2629	0.2001	0.1264	0.1271	0.1559
Simpson D’ = 1 − D	0.7371	0.7999	0.8736	0.8729	0.8441
Shannon H′	2.079	2.323	2.658	2.733	2.501
Evenness eH/S	0.1453	0.1594	0.2195	0.2365	0.1906

**Table 3 microorganisms-11-01326-t003:** Hematological parameters in broiler chickens after QC, VN and UF diet supplementation.

Parameters	Diet Supplementation Groups				
	BD	QC	QC + UF	VN	VN + UF
WBC, 10^9^ cell/L	49.60 ± 8.94	29.68 ± 2.10 *	35.73 ± 5.89 *	29.58 ± 0.47 *	27.23 ± 1.51 *
LYMP, %	53.63 ± 3.24	51.33 ± 5.19	56.23 ± 2.34	56.93 ± 2.16	51.33 ± 4.14
MONO, %	7.53 ± 0.41	7.83 ± 0.69	7.63 ± 0.68	7.40 ± 0.47	7.73 ± 0.90
GRAN, %	38.85 ± 3.10	40.85 ± 4.60	36.15 ± 1.87	35.68 ± 1.70	40.95 ± 3.72
RBC, 10^12^ cell/L	3.68 ± 0.09	3.79 ± 0.08	3.72 ± 0.17	3.32 ± 0.16	3.57 ± 0.21
HGB, g/L	113.25 ± 13.76	105.25 ± 1.65	103.50 ± 9.54	108.75 ± 7.82	109.00 ± 4.24
HCT, %	21.75 ± 3.19	19.38 ± 0.23	18.93 ± 1.85	19.83 ± 1.43	20.23 ± 1.04
MCV, fl	111.88 ± 3.11	109.35 ± 3.81	111.38 ± 2.49	109.40 ± 1.58	111.28 ± 1.78
MCH, pg	58.50 ± 0.65	59.18 ± 2.17	60.80 ± 0.74	59.30 ± 1.22	59.95 ± 1.55
MCHC, g/L	525.25 ± 11.50	542.75 ± 4.57	547.75 ± 8.85	552.50 ± 6.99	539.50 ± 8.70
RDW_CV, %	10.78 ± 0.47	10.95 ± 0.16	11.75 ± 1.32	10.88 ± 0.18	10.58 ± 0.31
RDW_SD, fl	37.38 ± 2.83	35.75 ± 0.65	40.63 ± 5.16	36.08 ± 0.62	36.08 ± 1.11
PLT, 10^9^ cell/L	91.00 ± 19.85	62.00 ± 5.28 *	61.75 ± 4.61 *	65.00 ± 5.58 *	74.00 ± 9.05
MPV, fl	19.03 ± 0.83	19.93 ± 0.73	20.68 ± 0.08	20.25 ± 0.26	19.75 ± 0.80
PCT, %	0.16 ± 0.03	0.12 ± 0.01 *	0.12 ± 0.01 *	0.13 ± 0.01 *	0.14 ± 0.01 *

Data presented as M ± SEM (were M—mean, SEM—standard error of sample means). *—*p* < 0.05 in comparisons with the control group.

## Data Availability

The sequencing raw data may be obtained upon request by e-mail at icis-ofrc@list.ru, belonging to the Institute for Cellular and Intracellular Symbiosis (ICIS) of the Ural Branch of the Russian Academy of Science (Orenburg, Russia).

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
