# Peer review of "Plant-Derived Quorum Sensing Inhibitors (Quercetin, Vanillin and Umbelliferon) Modulate Cecal Microbiome, Reduces Inflammation and Affect Production Efficiency in Broiler Chickens"

_microorganisms, 2023, doi:10.3390/microorganisms11051326_

Round 1
Reviewer 1 Report
The paper descibes the effects of plant-derived quorum sensing inhibitors on microbiome of chickens.
Taking into account the understanding of the impact of antibiotics additives in feeds on the spread of antibiotic resistance, this topic is very important for the the understanding how to improve the poultry production.
Overall the experimental design is relevant and idea is publication-worth.
Nevertheless, the some issues should be adressed before considering for publication.
Please specify from which gut part the samples were taken for microbiome sequencing
Please specify whether Fig 2, Tables 1,2 - are the values average/medians from several birds, provide SD or IQR
Is the any invormation about QC, VN and UF effects on eukariotic cells? What about impact on mucosa?
Is the any invormation about QC, VN and UF antibacterial effect? The observed changes in microbiota can be a subsequence of bacterial growth repression, not only QS-quenching. Th addition some data about biological effects of QC, VN and UF into introduction would be required.
Author Response
- Please specify from which gut part the samples were taken for microbiome sequencing.
The authors thank the reviewer for this remark, which makes the text more correct.
Samples for microbiome sequencing were taken from cecum part of chicken’s intestine due to their dense bacterial population, which aid in digestion of otherwise indigestible residues and is also a major reservoir for zoopathogens.
Line 126-130 revisited as:
After that the same birds were humanely euthanized, and the 200 µl each cecum contents were massaged into individual sterile Eppendorf tubes, containing 200 µl of DNA/RNA Shield (Zymo Research, USA), immediately frozen on dry ice, and stored at −80 °C until DNA extraction and following 16S-rRNA gene sequencing.
In addition, throughout the text, the terms "gut" and "gut microbiome" are paraphrased as "cecum” and “cecum microbiome".
- Please specify whether Fig 2, Tables 1,2 - are the values average/medians from several birds, provide SD or IQR.
Differences in the phyla and genera abundances in the control and experimental sub-groups (Fig 2, Table 1) were assessed by the Mann–Whitney (U) test.
This was done using the online calculator https://www.psychol-ok.ru/statistics/mann-whitney/mann-whitney_02.html. The null hypothesis was rejected at “p” values less than 0.05.
Due to the large number of observed values, which can make it difficult to read, we presented median data without IQR diapasons and ask keep the table in its original form.
- Is the any information about QC, VN and UF effects on eukaryotic cells? What about impact on mucosa? The addition some data about biological effects of QC, VN and UF into introduction would be required.
The authors agree with the reviewer that QC, VN and UF are not QSIs only, but in a more general concept they are “phytobiotics” with a variety of beneficial biological activities.
The antidiabetic, anti-inflammatory, antioxidant, antimicrobial, anti-Alzheimer’s, anti-arthritic, cardiovascular, and wound-healing effects of QC have been extensively investigated [Salehi, B.; Machin, L.; Monzote, L.; Sharifi-Rad, J.; Ezzat, S.M.; Salem, M.A.; Merghany, R.M.; El Mahdy, N.M.; Kılıç, C.S.; Sytar, O.; Sharifi-Rad, M.; Sharopov, F.; Martins, N.; Martorell, M.; Cho, W.C. Therapeutic Potential of Quercetin: New Insights and Perspectives for Human Health. ACS Omega. 2020, 5(20), 11849-11872. doi: 10.1021/acsomega.0c01818.].
Recently, bioactive properties of VN, such as neuroprotection, anti-carcinogenic, and antioxidant are gaining attention [Arya, S.S.; Rookes, J.E.; Cahill, D.M.; Lenka, S.K. Vanillin: a review on the therapeutic prospects of a popular flavouring molecule. ADV TRADIT MED (ADTM). 2021, 21(3), 1–17. doi: 10.1007/s13596-020-00531-w.].
UF shows antibacterial and antifungal activities, so as anti-inflammatory, anti-hyperglycaemic, molluscicidal and anti-tumor properties [Mazimba, O. Umbelliferone: Sources, chemistry and bioactivities review. Bulletin of Faculty of Pharmacy. 2017, 55(2), 223-232 doi:10.1016/j.bfopcu.2017.05.001.].
In the current literature, commercially available QC, VN, and UF used in a non-antibiotic strategy to improve the health and performance of poultry are generally classified as nature identical botanical compounds (NICs) with a wide spectrum of biological activity [Rossi, B.; Toschi, A.; Piva, A.; Grilli, E. Single components of botanicals and nature-identical compounds as a non- antibiotic strategy to ameliorate health status and improve performance in poultry and pigs. Nutr. Res. Rev. 2020, 33(2), 218-234. doi: 10.1017/S0954422420000013.].
Appropriate corrections are made in the "Introduction" section and the newly "Conclusion" section.
- Is the any information about QC, VN and UF antibacterial effect? The observed changes in microbiota can be a subsequence of bacterial growth repression, not only QS-quenching.
Indeed, QC, VN and UF show antibacterial activity (see references above), however, this effect develops only at very high doses. The doses used in the study are less than the minimum inhibitory concentrations, so authors are sure that they observe the QSI effect supplemented by the beneficial effects of QC, VN and UF on the host.

Reviewer 2 Report
Deryabin's study examined the impact of quercetin, vanillin, and umbelliferon on broiler chickens' gut microbiome, inflammation markers, and productivity. The findings demonstrated alterations in the microbiome, enhanced inflammatory status, and improved productivity. Nevertheless, the manuscript requires improvements before it can be considered for publication. The methodology requires further clarification, and there are significant issues with the manuscript's writing, including numerous grammatical errors.
Detailed points:
Line 50: please use more moderate wording – “any” is too strong. Some studies have found selective pressure.
Line 51-53: rewrite. Sentence is quite confusing.
Line 61: allowed to use? Problems with the grammar
Line 83: allowed to use? Problems with the grammar
Line 108: is this correct? 457 day-old?
Figure 1: Description of Figure 1 and study design in the text do not match.
Line 304: lacking a verb
Line 305: strange construction
Line 322: lacking words
Line 335: not be approved? Strange construction
Line 344: may could be?
Discussion in general: please discuss these results in terms of concentrations comparing your study with those in the literature.
The writing has too many errors to be identified effectively. Therefore, authors should have their manuscript reviewed by an individual with proficient knowledge in the language.
Author Response
Reviewer #2 comments:
Authors thank the reviewer for the identified errors and try to make the text understandable for English-speaking researchers.
Based on the detailed points, the following changes were made to the text:
Line 50: please use more moderate wording – “any” is too strong. Some studies have found selective pressure.
Sentence changed as follows:
“the quorum sensing has been evaluated as a promising antibacterial target where QSIs provide effective reduction of pathogenic behaviors without significant selective pressure against bacterial population”.
Line 51-53: rewrite. Sentence is quite confusing.
Revised version:
“Remarkable, natural QSIs are often plant-derived compounds found in edible or medicinal plants, which gives a chance for a rapid transition from scientific research to practical use in livestock as an alternative to AGPs”.
Line 61: allowed to use? Problems with the grammar.
Changed to “… approved for use…”.
Line 83: allowed to use? Problems with the grammar.
No error found
Line 108: is this correct? 457 day-old?
The sentence is rephrased as:
“450 Arbor Acres chicks (Aviagen LLC, Russia), which reached 7 days’ age at the start of the experiment, were randomly separated into 5 groups of 90 birds and then di-vided for tree trials of 30 animals each”.
Figure 1: Description of Figure 1 and study design in the text do not match.
The Figure legend is detailed, which makes it correspond to the text.
Line 304: lacking a verb
Revised version:
“The final broiler weight was 2736.50±215.4 g in the control group, which, taking in-to account the initial weight of 7-day-old chickens equal to 182.50 ± 8.1 g, gave an average growth per day of 72.97 g”.
Line 305: strange construction.
The sentence is rephrased as:
“Diet supplementation with QC and QC+UV did not change these zootechnical data, while in the VN group, the weight parameters tended to increase to 2816.00 ± 135.7 / 75.24 g, being significantly higher compared with the control after VN+UV supplementation (2847.00±160.4/76.13 g; p<0.05)”.
Line 322: lacking words.
No error found. In the current version, line 322 is the heading of the “Discussion” section.
Line 335: not be approved? Strange construction.
Revised version:
“…however, due to their absence in the microbiome, this hypothesis could not be verified, and the used Arbor Acres chickens was rated as specific pathogens free line”.
Line 344: may could be?
Thanks for the error detection, the word "may" has been deleted.
Discussion in general: please discuss these results in terms of concentrations comparing your study with those in the literature.
Authors plan to do this comparison as part of a subsequent meta-analysis in a separate study.
The manuscript under revision presents original data only.

Reviewer 3 Report
Dear editors and authors
The MS entitled ” Plant-Derived Quorum Sensing Inhibitors (Quercetin, Vanillin and Umbelliferon) Modulate Gut Microbiome, Reduces Inflammation and Affect Production Efficiency in Broiler Chickens” includes significant data about the effect of Quercetin, Vanillin and Umbelliferon on Gut Microbiome and potential anti-inflammatory activity.
Methods
Figure 1, as the figures are self-representing so all abbreviations on figure 1 need to be explained on the figure legend.
Results
-Figure 2, color on the figure mismatch and unrepresentative with the figure legends, no green color on the figure indicating Actinomycetota, No orange color for Mucoplasmatota.
- The effect on bacterial count in the colon need to be performed as Bacterila as Table 1; The values in the columns represent the relative abundances (%) of the specified taxon and did not indicate the bacterial count in the color
Minor
Line 108 need rephrase.
Dear editors and authors
The MS entitled ” Plant-Derived Quorum Sensing Inhibitors (Quercetin, Vanillin and Umbelliferon) Modulate Gut Microbiome, Reduces Inflammation and Affect Production Efficiency in Broiler Chickens” includes significant data about the effect of Quercetin, Vanillin and Umbelliferon on Gut Microbiome and potential anti-inflammatory activity.
Methods
Figure 1, as the figures are self-representing so all abbreviations on figure 1 need to be explained on the figure legend.
Results
-Figure 2, color on the figure mismatch and unrepresentative with the figure legends, no green color on the figure indicating Actinomycetota, No orange color for Mucoplasmatota.
- The effect on bacterial count in the colon need to be performed as Bacterila as Table 1; The values in the columns represent the relative abundances (%) of the specified taxon and did not indicate the bacterial count in the color
Minor
Line 108 need rephrase.
Author Response
Reviewer #3 comments:
- Figure 1, as the figures are self-representing so all abbreviations on figure 1 need to be explained on the figure legend.
Done
- Figure 2, color on the figure mismatch and unrepresentative with the figure legends, no green color on the figure indicating Actinomycetota, no orange color for Mucoplasmatota.
In accordance with the reviewer’s comments, we have corrected Figure 2 through plot scaling up to 0.1 and after 0.9, and currently, the colors indicating the Actinomycetota and Mucoplasmatota phylum have become distinguishable.
At the same time, it should be noted that Figure 2 clearly demonstrates the predominance of Bacillota and Bacteriodota phylum and the change in their ratio following diet supplementation, which is one of the important conclusions of the study.
- The effect on bacterial count in the colon need to be performed as Bacteria as Table 1; The values in the columns represent the relative abundances (%) of the specified taxon and did not indicate the bacterial count in the color.
The microbiome evaluation method used, based on 16S-rRNA gene sequencing, makes it possible to estimate the relative abundances of specified taxa, but not absolute bacterial count. This method is currently the most widely used for the microbiomes characterization, as it allows you to correctly evaluate the entire population of both cultivated and non-cultivated bacteria. Since no more than 5% of the total microbiome may be detected by bacterial counting on nutrient media, this method was not used in the study.
- Line 108 need rephrase.
This line rephrased as:
450 Arbor Acres chicks (Aviagen LLC, Russia), which reached 7 days’ age at the start of the experiment, were randomly separated into 5 groups of 90 birds and then divided for tree trials of 30 animals each.
All the revisions marked in the text by green, and a revised manuscript resubmitted.

Reviewer 4 Report
The manuscript under review covers modern areas of microbiological research, including microbial ecology and related aspects of immunology and productivity of farmed animals, which is in line with the current topics of the “Microorganisms” open access journal. This is an original research article presenting new data on plant-derived compounds as a potential alternative for antibiotic growth promoters, as evidenced by their beneficial effects on the broiler gut microbiome and following anti-inflammatory effect and increased poultry productivity. The article can be recommended for publication after making the changes indicated below: 1) The authors should explain why they limited their study of the gut microbiome to the analysis of the Bacteria domain, but did not include the Archaea domain in the study. According to current data, the Archaea domain is a typical inhabitant of the chicken gut microbiome [Saengkerdsub S., Herrera P., Woodward C.L., Anderson R.C., Nisbet D.J. and Ricke S.C. Identification and quantification of methanogenic archaea in adult chicken ceca. Appl. Environ. Microbiol. (2007) 73:353–6. doi:10.1128/AEM.01931-06]. 2) Microbiome research typically involves release of sequencing data (eg, by making available under accession number in the NCBI Sequence Read Archive https://www.ncbi.nlm.nih.gov/sra/ that provides a public repository for DNA sequencing data). Authors are invited to make such a placement or provide another way to obtain sequencing raw data. 3) The authors designate the studied compounds quercetin, vanillin and umbelliferone as quorum sensing inhibitors (QSIs), while in the available literature they are usually classified as nature-identical botanical compounds (NICs) showing a wider range of bioactivities [Rossi B., Toschi A., Piva A., Grilli E. Single components of botanicals and nature-identical compounds as a non- antibiotic strategy to ameliorate health status and improve performance in poultry and pigs. Nutr. Res. Rev. 2020 Dec;33(2):218-234. doi: 10.1017/S0954422420000013.]. Authors are invited to establish similarities and differences between QSIs and NICs concepts, stating this in the "Conclusion" section, which is a mandatory part of the article, but is missing in the peer- reviewed manuscript.Author Response
- The authors should explain why they limited their study of the gut microbiome to the analysis of the Bacteria domain, but did not include the Archaea domain in the study. According to current data, the Archaea domain is a typical inhabitant of the chicken gut microbiome
The study aims to find alternatives to antibiotic growth promoters that suppress zoopathogenic bacteria in the gut microbiome. Since all zoopathogenic taxa belong to the Bacteria domain and pathogens from the Archaea domain have not yet been described, the study was focused on the bacterial community only.
- Microbiome research typically involves release of sequencing data (eg, by making available under accession number in the NCBI Sequence Read Archive https://www.ncbi.nlm.nih.gov/sra/ that provides a public repository for DNA sequencing data). Authors are invited to make such a placement or provide another way to obtain sequencing raw data.
As reported in the “Acknowledgments” section, the chicken gut microbiomes analyses were performed in collaboration with the "Persistence of microorganisms" center of Collective Scientific Equipment at the Institute for Cellular and Intracellular Symbiosis (ICIS) of the Ural Branch of the Russian Academy of Science (Orenburg, Russia). The sequencing raw data may be obtained upon request by e-mail: icis-ofrc@list.ru, belonging to the ICIS. This note has been added to the “Acknowledgments” section.
- The authors designate the studied compounds quercetin, vanillin and umbelliferon as quorum sensing inhibitors (QSIs), while in the available literature they are usually classified as nature-identical botanical compounds (NICs) showing a wider range of bioactivities. Authors are invited to establish similarities and differences between QSIs and NICs concepts, stating this in the "Conclusion" section, which is a mandatory part of the article, but is missing in the peer- reviewed manuscript.
In accordance with the reviewer’s comments, the "Conclusion" section is introduced in the manuscript and the recommended article by Rossi et al. (2020) is included in the “References” list.
All the revisions marked in the text by green, and a revised manuscript resubmitted.
